# Efficient Convolutional Neural Network Training with Direct Feedback Alignment

## Abstract

There were many algorithms to substitute the back-propagation (BP) in the deep neural network (DNN) training. However, they could not become popular because their training accuracy and the computational efficiency were worse than BP. One of them was direct feedback alignment (DFA), but it showed low training performance especially for the convolutional neural network (CNN). In this paper, we overcome the limitation of the DFA algorithm by combining with the conventional BP during the CNN training. To improve the training stability, we also suggest the feedback weight initialization method by analyzing the patterns of the fixed random matrices in the DFA. Finally, we propose the new training algorithm, binary direct feedback alignment (BDFA) to minimize the computational cost while maintaining the training accuracy compared with the DFA. In our experiments, we use the CIFAR-10 and CIFAR-100 dataset to simulate the CNN learning from the scratch and apply the BDFA to the online learning based object tracking application to examine the training in the small dataset environment. Our proposed algorithms show better performance than conventional BP in both two different training tasks especially when the dataset is small.

## 1 Introduction

Deep learning becomes the core of the machine learning and it has been utilized for many applications such as machine translation (Singh et al. (2017)), speech recognition (Wang et al. (2017)), and object classification (Rawat & Wang (2017)). Training the deep neural network (DNN) is an important portion of the deep learning because we need different pre-trained models to cover the various deep learning tasks. One well-known method of DNN training is the algorithm called back-propagation (BP) (Rumelhart et al. (1986)). The BP is the gradient descent based training method which follows the steepest gradient to find the optimum weights. Therefore, BP based training can be applied to any DNN configurations if the network consists of any differentiable operations. For instance, not only multi-layer perceptron (MLP) but also both convolutional neural network (CNN, LeCun et al. (1998)) and recurrent neural network (RNN, Hopfield (1982)) can be trained by using the BP.

Even though the BP shows the outstanding performance in DNN training, BP based training suffers from the overfitting problem. Since the BP easily sinks into a local minimum, we need a large scale of the dataset to avoid the overfitting. In addition, we use various data augmentation techniques such as flipping and cropping. If the DNN training is done in the limited resources and dataset, BP based DNN training is too slow to be converged and shows low accuracy. One example is MDNet (Nam & Han (2016)) which introduces an object tracking algorithm with the online learning concept. Real-time implementation is an important issue in object tracking, so it has the limitation to utilize various data augmentation and large dataset for online learning.

To break the BP based DNN training paradigm, a lot of new training methods have been developed. Salimans et al. (2017) proposed the evolution strategy which searches various weight without the gradient descent. It has a chance to be computed in parallel, but it needs a lot of seeds to find the global optimal solution. Moreover, evolution strategy has a slow convergence problem which can be an obstacle for fast online learning application. Another algorithm, feedback alignment (FA, Lillicrap et al. (2014)), was proposed based on the gradient descent methodology, but without following the steepest gradient. The FA pre-defines a feedback weight before starting the training,

and the weight is determined by the random values.[1] It is also known to converge slowly, and shows worse performance compared with the previous BP.

Direct feedback alignment (DFA, Nokland (2016)) was developed by getting the idea from the FA. Although the FA propagates errors from the last layer back to the first layer step-by-step, the DFA propagates errors from the last layer directly to each layer. This approach gives the opportunity of the parallel processing. Since the errors of every layer are generated independently in DFA, we can immediately calculate each layer's gradient if the DNN inference is finished. Moreover, the number of required computation is reduced in the DFA. This is because the number of neurons in the last layer is usually fewer than the prior layer, so the size of the feedback weight becomes smaller than the BP. In spite of these advantages, the DFA suffers from accuracy degradation problem. The accuracy degradation problem becomes more serious when the DFA is applied to the CNN case. It is known that the DNN is not learnable with the DFA if the DNN becomes much deeper than AlexNet (Krizhevsky et al. (2012)).

In this paper, we explain the new DFA algorithm to improve the training accuracy in CNN, and suggest a feedback weight initialization method for the fast convergence. Moreover, we propose the binary direct feedback alignment (BDFA) to maximize the computational efficiency. We verified the training performance in the VGG-16 (Simonyan & Zisserman (2015)) without any data augmentation, and DFA shows higher accuracy compared with the BP. And then, the training with small dataset was proved through the online learning based object tracking application. Our proposed algorithm shows better performance than the conventional BP approach in both the learning from the scratch and the object tracking tasks.

The remaining part of the paper is organized as follows. The mathematical notation of the BP, FA, and DFA will be introduced in Section 2. Then, the details of the proposed algorithms will be explained in Section 3. The experiment will be followed in Section 4 and the paper will be concluded in Section 5.

## 2 PRELIMINARIES

### 2.1 BACK-PROPAGATION

Back-propagation is a general algorithm for the DNN training, suggested by Rumelhart et al. (1986). Let $\boldsymbol{o}_i$ be the $i^{th}$ layer's feature map, when $\boldsymbol{W}_{i+1,i}$ be the weight and bias between the $i^{th}$ layer and the $i+1^{th}$ layer. If the activation function is represented as $f(\cdot)$, and $L$ is the total number of layers, the feature map of the $i+1^{th}$ layer can be calculated as

$$\boldsymbol{i}_{i+1} = \boldsymbol{W}_{i+1,i}\,\boldsymbol{o}_i, \quad \boldsymbol{o}_{i+1} = f(\boldsymbol{i}_{i+1}), \quad i \in \{0,1,\ldots,L-1\} \tag{1}$$

Various activation functions such as sigmoid, tanh, and ReLU (Nair & Hinton (2010)) can be one of candidates of the $f(\cdot)$. Some DNNs such as generative adversarial networks (Goodfellow et al. (2014)) use more advanced activation functions like Leaky ReLU and PReLU (Xu et al. (2015)). Once the inference is over, the inference result is compared with the pre-defined labels, and the error map $\boldsymbol{e}_L$ is calculated by a loss function such as cross-entropy. The error is propagated gradually from the last layer to the first layer, and the $i^{th}$ layer's error map can be calculated as

$$\boldsymbol{e}_i = (\boldsymbol{W}_{i+1,i}^T\,\boldsymbol{e}_{i+1}) \odot f'(\boldsymbol{i}_{i+1}), \quad i \in \{1,2,\ldots,L-1\} \tag{2}$$

where $\odot$ is an element-wise multiplication operator and $f'()$ is the derivative of the non-linear function. We need the transposed weight matrix, $\boldsymbol{W}_{i+1,i}^T$, to propagate the errors in MLP case. After the BP, the gradient of each layer is computed by using both the $i^{th}$ layer's feature map and $i+1^{th}$ layer's error map. The $i^{th}$ layer's gradient, $\boldsymbol{G}_i$ is calculated as following.

$$\boldsymbol{G}_{i,b} = \boldsymbol{e}_{i+1}\,\boldsymbol{o}_i^T, \quad i \in \{0,1,\ldots,L-1\}, \quad b \in \{0,1,\ldots,B\} \tag{3}$$

$$\boldsymbol{G}_i = \frac{1}{B}\sum_b \boldsymbol{G}_{i,b}, \quad \boldsymbol{W}'_{i+1,i} = \boldsymbol{W}_{i+1,i} - \eta\,\boldsymbol{G}_i \tag{4}$$

---

[1]The feedback weight was notated as a fixed random matrix in the original paper

If we use the mini-batch gradient descent with the batch size $B$, total $B$ gradients, $\boldsymbol{G}_{i,b}$ are averaged and get the $\boldsymbol{G}_i$. $\boldsymbol{G}_i$ is used to update the $\boldsymbol{W}_{i+1,i}$ by multiplying learning rate, $\eta$. In the CNN case, the matrix multiplication operations in every step described in MLP is substituted with the convolution operations. One more different thing is that it uses 180°flipped kernel instead of the transposed weight used in the BP. Other operations are as same as in the MLP case.

## 2.2 FEEDBACK ALIGNMENT

Feedback alignment (FA) which was introduced by Lillicrap et al. (2016) substituted the transposed weight with the fixed random matrix in error propagation operation. In other words, they pre-defined the feedback weight which has the same size as the transposed weight, but the values of the feedback weight are determined randomly. Although the weights used in the training are updated for every iteration, the pre-defined random matrix is maintained until the training is finished. Let $\boldsymbol{R}_{i+1,i}$ be the feedback weight of the $i^{th}$ layer, the propagated error in the FA is calculated as

$$\boldsymbol{e}_i = (\boldsymbol{R}_{i+1,i}^T \, \boldsymbol{e}_{i+1}) \odot f'(\boldsymbol{i}_{i+1}) \tag{5}$$

, and the other procedures such as gradient generation and weight updating are same as BP does.

## 2.3 DIRECT FEEDBACK ALIGNMENT

Even though, both the BP and the FA propagate errors from the last layer to the first layer in order, the DFA (Nokland (2016)) directly propagates errors from the last layer to other layers. If the number of neurons in the last layer and the $i^{th}$ layer, are represented by $N_L$ and $N_i$ respectively, the size of the feedback weight is determined as $N_i \times N_L$. Let, $i^{th}$ layer's feedback weight in the DFA be $\boldsymbol{D}_i^T$, then the error is calculated as

$$\boldsymbol{e}_i = (\boldsymbol{D}_i^T \, \boldsymbol{e}_L) \odot f'(\boldsymbol{i}_{i+1}) \tag{6}$$

One of the interesting characteristics is that there is no data dependency between different errors because the DFA propagates the errors directly from the last layer. This characteristic gives the opportunity of parallel processing in the DFA based error propagation operation.

Compared with the BP, DFA showed similar training performance for the multi-layer perceptron (MLP) structure. Moreover, DFA requires fewer computations because $N_L$ is generally smaller than the number of neurons in the intermediate layers. However, the DFA dramatically degrades the accuracy when it applied in the CNN training as shown in Figure 1. Furthermore, the elements of the intermediate feature map are all connected with the last layer neurons, so it requires much more computations compared with BP based CNN training. In summary, the DFA's computational efficiency can be induced in the MLP training, but not in the CNN.

## 3 OUR APPROACH

As mentioned in Section 2, the DFA is efficient for MLP training, because it can be computed in parallel without accuracy degradation. However, this advantage is diminished when the DFA is applied in the CNN. To solve this problem, we propose the new training method to make the DFA applied

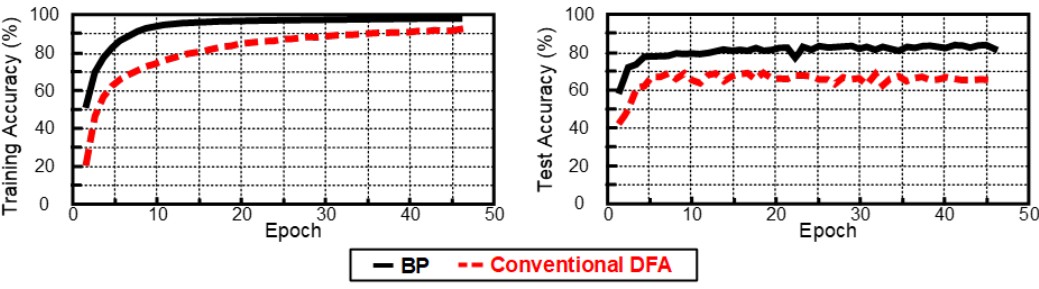

Figure 1: BP (Black) vs Conventional DFA (Read) for CNN Training (Tested in CIFAR-10)

to the CNN. In addition, the initialization method of the feedback weight is suggested for the fast and stable learning curve. At last, we propose the advanced training algorithm, binary direct feedback alignment (BDFA), which shows high computing efficiency and robust training performance in various conditions.

## 3.1 CDFA: CNN TRAINING BY COMBINING BOTH BP AND DFA

Generally, CNN consists of the convolutional layers and fully-connected (FC) layers. The two different kinds of layers have different roles. For example in object classification, convolutional layers are considered as the feature extractor by using the characteristics of the convolution computation. In contrast, FC layers receive the result of the feature extractor and judge what the object is. However, in the initial iterations of the BP based CNN training, training the convolutional layers can be ambiguous because the FC layers cannot be considered as a good object classifier. During some iterations of BP, the weight of the FC layers will be changed and the convolutional layers should be adaptive to the changed FC layers. In other words, convolutional layers can be confused if the propagated error's domain is changed for every iteration.

If the error is propagated from the last FC layer to convolutional layer through the constant domain shifting, the training of the convolutional layer can be more stable than conventional BP. From this motivation, we use DFA instead of BP in the FC layers. As shown in Figure 2, the network maintains the BP in the convolutional layers but adopts the DFA for the FC layers. Since there is no data dependency among the FC layers, the DFA can directly propagate the error to the FC1. Since the DFA uses fixed feedback weight for error propagation, convolutional layers do not have to be adaptive to various errors which are derived from the different domain. In this method, the error propagation in the convolutional layers can be started even though the errors are not propagated for the remained FC layers. Therefore, the error propagation of both the convolutional layers and the FC layers can be computed in parallel right after DFA is done for the first FC layer.

Nokland (2016) shows that the randomly initialized feedback weight can be used for DFA based DNN training. However, both the FA and the DFA are sensitive to the initialization method of the feedback weight because it affects the training accuracy and the convergence speed significantly. As shown in Liao et al. (2015), it is observed that the batch-normalization (BN) seems to make the FA become not sensitive to the initialization method, but it is still a problem because of the slow convergence. To make the DFA robust to the initialization method, we fixed the feedback weight as the multiplication of the transposed weights in multiple layers. To sum up, the feedback weight of the $i^{th}$ layer, $\boldsymbol{D}_i$, can be calculated as

$$\boldsymbol{D}_i = \boldsymbol{W}_{L,L-1} \ldots \boldsymbol{W}_{i+2,i+1} \boldsymbol{W}_{i+1,i} \tag{7}$$

, and finally, the error propagation operation can be summarized as follow.

$$\boldsymbol{e}_i = (\boldsymbol{D}_i^T \boldsymbol{e}_L) \odot f'(\boldsymbol{i}_{i+1}) = (\boldsymbol{W}_{i+1,i}^T \ldots \boldsymbol{W}_{L-1,L-2}^T \boldsymbol{W}_{L,L-1}^T \boldsymbol{e}_L) \odot f'(\boldsymbol{i}_{i+1}) \tag{8}$$

The suggested initialization method is suitable for other various functions such as sigmoid, tanh and ReLU. Moreover, other normalization or optimization methods such as BN and dropout (Srivastava et al. (2014)) are also applicable with the proposed initialization with the equation (7).

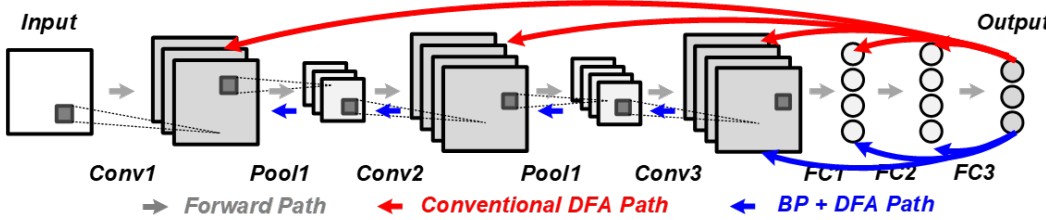

Figure 2: Conventional DFA in CNN and Proposed DFA based Error Propagation

### 3.2 BINARY DIRECT FEEDBACK ALIGNMENT (BDFA)

The DFA needs the feedback weight addition to the forward weight matrix, and it occupies a larger memory to store both two matrices. Moreover, the DFA has the chance to be computed in parallel, but it requires much larger memory bandwidth. Since the throughput in the FC computing is vulnerable to the memory bandwidth, loading the additional feedback weight degrades the throughput compared with the unlimited bandwidth case.

To solve the throughput bottleneck problem caused limited bandwidth, we propose the binary direct feedback alignment (BDFA) algorithm. BDFA uses the binarized feedback weight, $B_i$, whose values are determined as either +1 or -1. In other words, the $B_i$ can be stored as a single bit to represent only the sign value of the feedback weight's element. As a result, required memory to store the $B_i$ is reduced by 96.9% compared with the 32-bit floating point representation which now becomes the general numeric representation in CPU or GPU. As we determined in the DFA, BDFA's feedback weight, $B_i$ can be similarly defined by a modification of the equation (7). $B_i$ is determined as

$$B_i = sign(W_{L,L-1} \ldots W_{i+2,i+1} W_{i+1,i}), \quad e_i = (B_i^T e_L) \odot f'(i_{i+1}) \qquad (9)$$

, when the $sign(\cdot)$ is the function which indicates the sign value of each element. The difference between equation (7) and (9) is only whether the $sign(\cdot)$ is applied or not. By applying equation (9), BDFA shows faster and stable training convergence compared with the random initialization case. The effect of the binarization and initialization will be discussed in section 4.

## 4 EXPERIMENTS

In this section, we compared the training accuracy of the conventional BP and suggested training algorithm. We measured the relative accuracy by training CNN from the scratch in CIFAR-10 and CIFAR-100 dataset (Krizhevsky (2009)). We used the network configuration as described in Figure 3. The base network follows the VGG-16 (Simonyan & Zisserman (2015)) configuration, but has one additional FC layer. To sum up, it consists of 13 convolutional layers with BN and ReLU activation functions, followed by three FC layers without BN. The number of neurons in the last FC layer is determined by the number of classification categories in each different dataset. In the BP based approach, both the convolutional layers and the FC layers are trained by using BP. In contrast, the training method of the last three FC layers is substituted with the DFA or the BDFA to measure the performance of the proposed training algorithm. The simulation was based on mini-batch gradient descent with the batch size 100 and uses momentum(Sutskever et al. (2013)) for the optimization method. The parameters of the network are initialized as introduced by He et al. (2015), and the learning rate decay and the weight decay method is adopted. Other hyper parameters are not changed for fair comparison.

### 4.1 CNN TRAINING FROM THE SCRATCH

The CNN training with the DFA and BDFA, are renamed as CDFA and CBDFA respectively for the simple explanation. Table 1 shows Top5 and Top1 test accuracy after the CNN training is done in the two different CIFAR datasets. In this simulation, there is no data augmentation to make an environment which has a limited dataset. This condition can examine whether the algorithm is robust to the training in the small dataset. To sum up, only 50,000 images in the CIFAR-10 and CIFAR-100 is only used for DNN training and the other 10,000 images are tested to evaluate the test accuracy. As a result, both the CDFA and the CBDFA show higher test accuracy compared with the

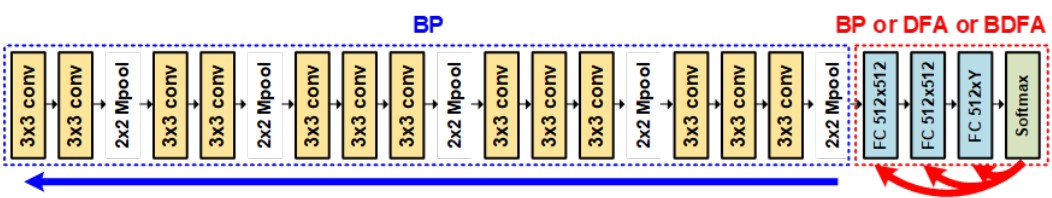

Figure 3: Overall network configuration for training

Table 1: CNN Training Result in CIFAR-10 & CIFAR-100 (Small Learning Rate)

| CIFAR 10 | BP | BP w/ BN | CDFA Random | CDFA Eq (7) | CDFA w/ BN | CBDFA Random | CBDFA Eq (9) | CBDFA w/ BN |
|---|---|---|---|---|---|---|---|---|
| Top5 | 98.63 | 98.24 | 98.42 | 98.55 | 98.56 | **98.63** | **98.88** | **98.83** |
| Top1 | 81.11 | 76.91 | **88.68** | 86.36 | 87.41 | **89.39** | **87.65** | 86.46 |

| CIFAR 100 | BP | BP w/ BN | CDFA Random | CDFA Eq (7) | CDFA w/ BN | CBDFA Random | CBDFA Eq (9) | CBDFA w/ BN |
|---|---|---|---|---|---|---|---|---|
| Top5 | 67.80 | 63.91 | **77.05** | 72.82 | **77.55** | 75.07 | 71.92 | **76.85** |
| Top1 | 40.29 | 37.80 | **61.42** | 48.24 | **55.11** | **59.92** | 47.48 | 54.47 |

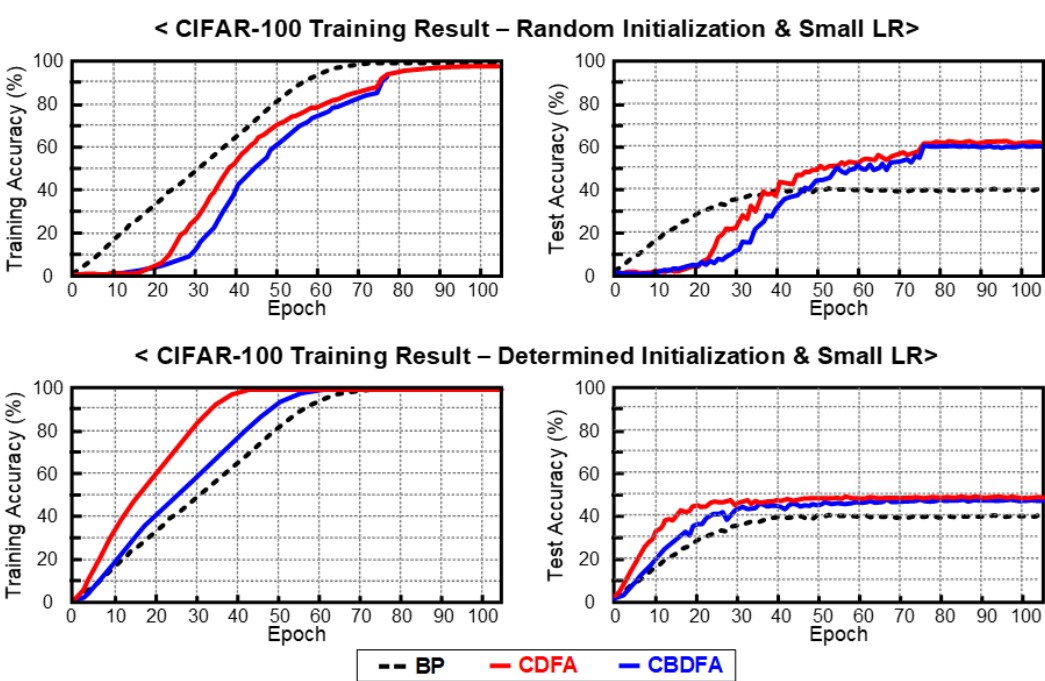

Figure 4: Training and Test Accuracy with Proposed Training Algorithm

conventional BP even though the feedback weight is randomly initialized. In CIFAR-10, the CDFA and CBDFA are 7.5% and 8.3% higher in Top1 test accuracy than the BP respectively. The accuracy improvement by the CDFA and the CBDFA seems much more remarkable in CIFAR-100. As shown in Figure 4, the training curve of the CDFA and CBDFA is much slower, but they achieve 21.3% and 19.6% better performance respectively compared with the BP. However, the feedback weight with the random initialization has critical problems for training. One of the problems is the slow training curve described in Figure 4. DFA requires time to be adaptive to the randomly initialized feedback weights, so it takes a long latency to be converged. In the BP approach, we generally take the larger learning rate to make the training faster. However, the test accuracy of the DFA and BDFA is swung up and down dramatically when the large learning rate is applied. Moreover, it still spends a long time to converge. In this problem, the initialization with the equation (7) and (9) can be useful to solve the learning speed and stability problem. After the feedback weight is initialized by the proposed equations, it shows faster and more stable convergence characteristic as shown in Figure 5. When the proposed initialization method is combined with the large learning rate, it shows the best training performance compared with the other results.

When the dataset is augmented with the flipping and cropping, the training performance of the BP and CBDFA becomes higher than before. The simulation uses the large learning rate, and CBDFA

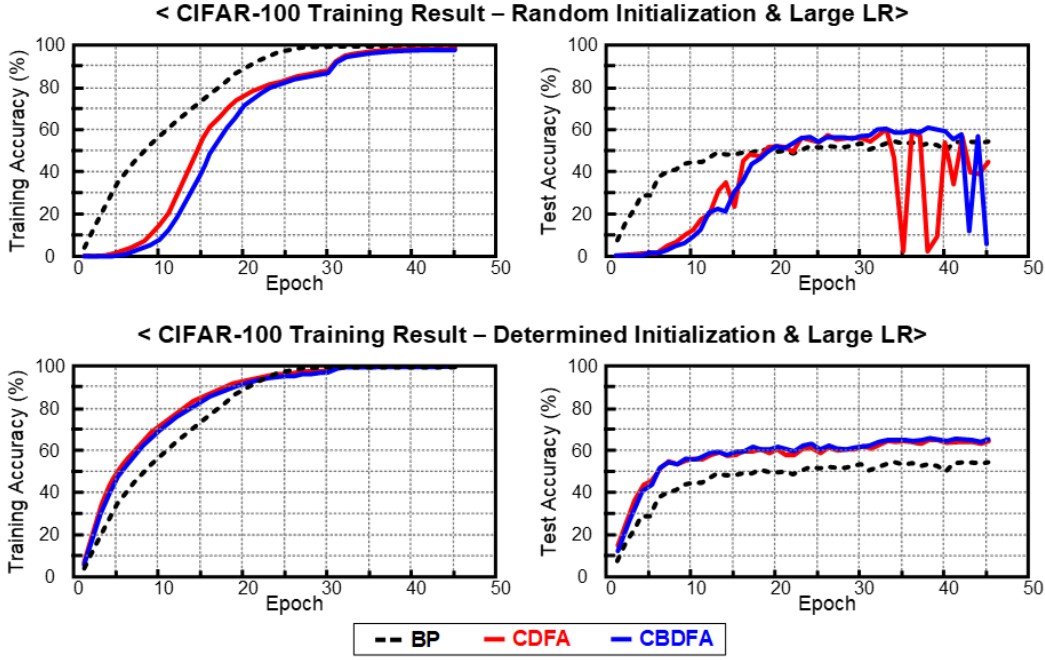

Figure 5: Training and Test Accuracy with Proposed Training Algorithm

Table 2: CNN Training Result with Data Augmentation (CIFAR-10)

|  | w/o Data Augmentation | | | w/ Data Augmentation | | |
|---|---|---|---|---|---|---|
|  | **BP** | **CBDFA** | **Conv. only Training** | **BP** | **CBDFA** | **Conv. only Training** |
| **Top5** | 99.15 | **99.07** | 98.84 | 99.33 | **99.49** | 99.46 |
| **Top1** | 82.33 | **87.35** | 82.06 | 87.97 | **90.13** | 88.48 |

takes the initialization with equation (9). In table 2, the performance of the CBDFA shows the highest accuracy compared with not only BP but also the training suggested by Hoffer et al. (2018). It trains only the convolutional layers, and the parameters of the FC layers are fixed. Even considering the data augmentation, CBDFA still shows higher training accuracy compared with the other two methods. As a result, CBDFA seems robust to the size of the dataset.

There are some interesting observations in our simulation. First of all, the CBDFA shows negligible accuracy degradation compared with the CDFA based training. Sometimes, the CBDFA has a rather better performance than the CDFA case. Refer to the learning curve described in both Figure 4 and Figure 5, the CBDFA's learning speed is slightly degraded, but the final training results are approximately the same. Therefore, CBDFA can improve the training performance and take the hardware benefits such as smaller memory bandwidth by adopting binarized feedback weights. The second one is the effect of the BN. Even though the suggested initialization method achieves the fast and stable training, it has a little accuracy degradation compared with the random initialization method. This accuracy degradation can be reduced when the BN layer is added after the FC layer. This result is counter characteristic compared with the BP case because the BP shows the worse training result when the BN layer is followed right after the FC layer. To sum up, the equation (7) and (9) are much more powerful when the BN is followed after the convolutional layer.

## 4.2 EXAMPLE OF ONLINE LEARNING WITH SMALL DATASET: OBJECT TRACKING

Nam & Han (2016) suggested online FC learning based object tracking algorithm, MDNet. The online learning concept in MDNet has derived many other algorithms such as BranchOut (Han et al. (2017)) and ADNet (Yun et al. (2017)). In the MDNet, both the convolutional layers and FC layers

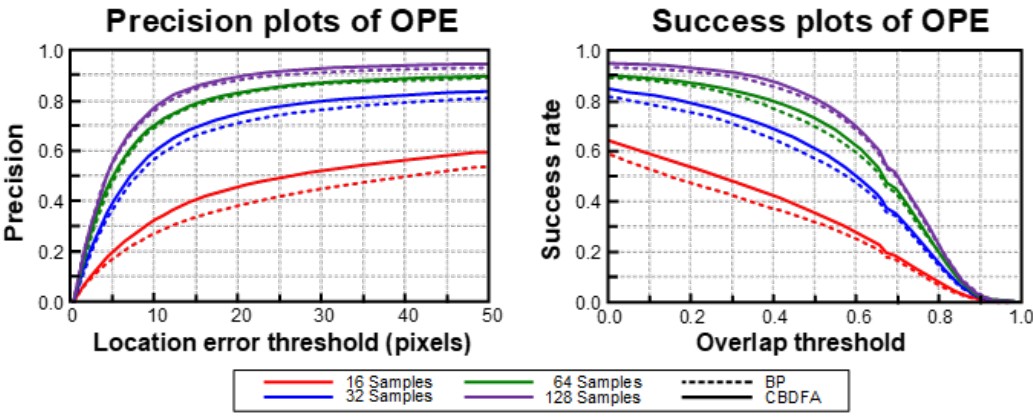

Figure 6: Object Tracking Result with the Proposed Training Algorithm, CBDFA

are pre-trained with the VOT (Kristan et al. (2013), Kristan et al. (2016)) and OTB (Wu et al. (2015)) object tracking dataset. However, the last layer of the FC layers is randomly initialized for the new tracking task. The convolutional layers do not need to be trained during the tracking but the FC layers are fine-tuned by using the BP. To apply BDFA to FC online learning with the small dataset, we replace the BP by the BDFA for FC layers in the MDNet.

We compared the object tracking performance by drawing the precision and the success plots of the one-pass evaluation (OPE) in the OTB dataset. As shown in Figure 6, the object tracking with the BDFA based online learning shows similar performance compared with the BP case. However, the BDFA shows better performance than BP when the batch size becomes smaller. Since the object tracking application is very sensitive to the online learning speed and the BDFA has a chance to be computed in parallel, BDFA based online learning becomes much more beneficial than conventional BP. Moreover, the BDFA can dramatically reduce the required data transaction in error propagation because of the fewer neuron interconnections and the binarization. As pointed out in the paper, Han et al. (2018), BP based online learning is inefficient for online learning in the devices which have limited memory bandwidth, computing resources, and small dataset. In this case, the effect of the BDFA can be maximized because of its profits.

## 5 CONCLUSION

In this work, we propose the new DNN learning algorithms to maximize the computation efficiency without accuracy degradation. We adopt one of the training method, DFA and combine it with the conventional BP. The combination of the DFA and the BP shows much better test accuracy in CNN training through the simulation in the CIFAFR-10 and CIFAR-100 dataset. BDFA takes one step further, binarizing the feedback weight while maintaining a similar performance compared with the full-precision DFA. The stability problem induced in the DFA and BDFA simulation (Figure 5) can be solved by the new feedback weight initialization method, equation (7) and (9). The BDFA is also simulated in the object tracking application, and it shows better tracking results compared with the conventional BP based online FC tuning.

In this research, we can see that the DFA can reduce the computational complexity and achieve better training performance than conventional training method. However, the feedback path of the DFA cannot still be applied directly into convolutional layer because of the significant accuracy degradation. To break the limitation of the current research, We will continue the research about expanding the usage of the DFA to not only convolutional layers but also other RNN networks such as LSTM (Hochreiter & Schmidhubers (1997)) and GRU (Chung et al. (2014)).

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
