# OpenReview forum: "Efficient Convolutional Neural Network Training with Direct Feedback Alignment"
_ICLR.cc/2019/Conference_

### Official Review · AnonReviewer3 · 2018-10-29
**The method needs more insight or novelty, and the results have room to improve**

**Rating:** 5
**Confidence:** 3

**Review:**

This manuscript extends the direct feedback alignment (DFA) approach to convolutional neural networks (CNN) by (1) only applying DFA to FC layers with backpropagation (BP) in place for convolutional layers (2) using binary numbers for feedback matrix.

Originality wise, I think (1) is a very straightforward extension to the original DFA approach by just applying DFA to places where it works. It still does not solve the ineffectiveness of DFA on convolutional layers. And there is no much insight obtained. (2) is interesting in that a binary matrix is sufficient to get good performance empirically. This would indeed save memory bandwidth and storage. This falls into the category of quantization or binarization, which is not super novel in the area of model compression.

The experimental results show that the proposed approach is better than BP based on accuracy. However, these results might be called into question because the shown accuracies on CIFAR10 and CIFAR100 are not state-of-the-art results. For example, the top 1 accuracy of CIFAR10 in this paper 81.11%. But with proper tuning, a CNN should be able to get more than 90% accuracy. See this page for more details.
http://rodrigob.github.io/are_we_there_yet/build/classification_datasets_results.html
Therefore, though the claimed accuracy of the proposed method is 89%, it is still not the state-of-the-art result and it seems to be lack of tuning for the BP approach to perform similar level of accuracy. The same conclusion applies to CIFAR100. In fact, from figure 4, the training accuracy gets 100% while the testing accuracy is around 40% for BP, which seems to be overfitting. With these results, it is hard to judge the significance of the manuscript.

Minor typos:
In Equation 1, the letter i is overloaded.

---

> ### Author Response · Authors · 2018-11-13
> **Thank you for your comments**
>
> Thank you for your comments.
>
> Followings are our response about your comments.
>
> 1) Effectiveness of DFA
> Reduced amount of computation by BDFA can be much smaller when we include the computations due to the covolutional layers. Therefore, computational efficiency caused by BDFA can be larger only for the specific NN models. For example, MDNet (referred in the paper) uses only FC fine-tuning, and BDFA can improve the computational efficiency significantly.
> However, when the BDFA is applied instead of BP, we can reduce the required size of the dataset to achieve same test accuracy. Finally, training with the small dataset can additionally improve the effectiveness of the BDFA based training.
>
> 2) Limited Novelty
> As you pointed out, we suggest the combination of two well-known training methods, BP and binarized DFA. Although the proposed solution is simple, the effect of the training is powerful especially when the training dataset is small. To emphasize the effect of the algorithm, we added the simulation of data augmentation (table 2).
>
> 3) Low baseline performance
> There are three reasons why the baseline model in the manuscript shows lower accuracy compared with the state-of-the-art CIFAR results.
>   (1) There was no data augmentation.
>   (2) Simple optimization, momentum is used.
>   (3) We used one additional FC layer for VGG-16.
> We added the simulation result with data augmentation in table 2. The explanation of the different network configuration is added in section 4.

---

### Official Review · AnonReviewer2 · 2018-10-29

**Rating:** 4
**Confidence:** 4

**Review:**

The paper propses to use a combination of Direct Feedback Allignment (DFA) and BackPropagation (BP) to improve upon standard back propagation.
To understand what is done, consider the following: Feedback Alignment is +- equal to back propagation when using random but fixed weights in the backwards pass. Direct feedback alignment uses random backprojections directly to the layer of interest.
The advantage of DFA is that It bypasses the normal computational graph. The advantage of this is that if compute is infinite, all of these updates can be computed in parallel instead of pipelining them as is done in standard BP.

In the current paper, the use of DFA for dense layers and BP for conv layers which is named CDFA is proposed.
In addition the paper also proposes a binarized version of BDFA to limit memory consumption and communication. It is claimed that the proposed techniques improve upon standard back propagation.

Overall, the paper is easy to understand, but I lean towards rejecting this paper because I am not convinced by the experimental evidence. As outlined below, the key issue is that the baseline appears to be weak. Additionally, the main limitation of the proposed approach can only benefit a very limited set of architectures.

Positive points:
---------------------
The authors did an excellent job of introducing BP, FA and DFA in the paper. This makes the core concepts and ideas accessable without having to delve through prior work.


The own contributions and the key idea is easy to understand.

Limitations and possible improvements
-------------------------------------------------------
A core limitation is that recent networks do not have a combination of dense layers and convolutional layers. In many cases the networks are fully convolutional, this limits the applicability of the proposed combination of DFA and BP. The use of additional networks would benefit the paper. Currently only VGG 16 on Cifar 10 is used. Also, the data augmentation strategy is not discussed. Of course, it would be nice if additional datasets could be included as well, but this of course depends on the computational resources the authors have available.


The key issue to me is that performance improvements for CIFAR are reported, but I fear that the baseline accuracy for VGG16 might be a bit low. If I memory serves me well, it should be able to achieve around 90% at least on CIFAR 10 using VGG style networks. I did a quick search and found http://torch.ch/blog/2015/07/30/cifar.html corroborating this but I did not verify this directly.


Related to the previous point, since this is an empirical paper, describing the hyper-parameter optimizations and final settings in detail  can convince the reader that the study is exectued correctly. Much of the information is missing now.


Similarly, I have trouble understanding section 4.1 and section 4.2 since I do not know the exact details of the experiments. This can be fixed easily however.


Provide complexity estimates of the potential speedup or provide actual timing information. (Although this might not be that meaningful without much additional work given that gpu kernels are often heavily optimized).


Last year there was a submission to ICLR about fixing the final output layer and only learning the convolutional layers. If we consider that random projections work remarkably well and can be considered approximations of kernels, it could be interesting to add a baseline where the fully connected layers are fixed and only the convolutional layers are trained. The error signal can be propagated using standard BP, FA or DFA methods but it would shed light on whether learning in the higher layers is actually needed or BP in the conv layers is sufficient.

Minor possible improvements
------------------------------------------
Finally, I would strongly suggest that the authors perform some additional proofreading. There are quite a few strange formulations and spelling mistakes. That being said, it did not prevent me from understanding the manuscript so this remark DID NOT factor into my judgement.

In addition to remark above, I would suggest removing the second paragraph from the introduction. It feels out of place to me, and the vanishing gradient effects are not discussed in the remainder of the manuscript.


The list of possible optimizers before the selection for SGD+Momentum is not needed. Simply stating that SGD with momentum is used should be sufficient.


“Training from scratch” instead of “Training from the scratch”

---

> ### Author Response · Authors · 2018-11-13
> **Response about your valuable comments**
>
> Thanks to your valuable comments, we can improve the quality of the manuscript more than before.
>
> Before answering your main concerns, the manuscripts are revised as follow.
>
> 1) The contents of the 2nd paragraph was gradient vanishing problem but it is replaced with the overfitting problem to emphasize the effect of the CDFA.
>
> 2) We made the figures smaller to secure more manuscript spaces.
>
> 3) Minor typos in the manuscripts were revised.
>
> ----------------------------------------
>
> Followings are our response about your comments.
>
> 1) Data augmentation strategy and the details of the experiment (hyper parameter, network configuration, data augmentation, optimization method, etc)
> As you pointed out, the data augmentation startegy was not mentioned in the previous manuscript. We added the explantion about the data augmentaion clearly and added more details of the experiments as follow.
>   - The base network follows the VGG-16 configuration, but has one additional FC layer.
>   - The simulation was based on mini-batch gradient descent with the batch size 100 and uses momentum for the optimization method.
>   - The parameters of the network is initialized as introduced by He et al. (2015), and the learning rate decay and the weight decay method is adopted. Other hyper parameters are not changed for fair comparison.
>
> In this experiments result in the previous manuscripts was evaluated with no data augmentation. Even though the data augmentation is not used, CBDFA shows more robust training performance compared with the BP. We emphasized this strength into the manuscript. Moreover, the related simulation is added in table 2.
>
> 2) Complexity and speedup estimation
> The complexity of the BP training highly depends on the network configurations. Instead, we add the reference, Han et al (ISCAS 2018) which explains the BP based online learning can be the obstacle of real-time object tracking implementation.
>
> 3) Comparison with "Conv. only training"
> In table 2, we added the comparison results with conv. only training suggested by Hoffer et al. (2018). CBDFA shows much better training performance compared with the conv. only training.
>
> 4) Explanation of gradient vanishing problem in intorudction.
> We removed the second paragraph, but add the overfitting problem.
>
> 5) Low baseline performance
> There are three reasons why the baseline model in the manuscript shows lower accuracy compared with the state-of-the-art CIFAR results.
>   (1) There was no data augmentation.
>   (2) Simple optimization, momentum is used.
>   (3) We used one additional FC layer for VGG-16.
> We added the simulation result with data augmentation in table 2. The explanation of the different network configuration is added in section 4.

---

> > ### Comment · AnonReviewer2 · 2018-12-05
> > **Final comments**
> >
> > I am impressed by how quickly the authors addressed some of the issues in the paper.
> >
> > Despite this, I feel that the method has limited applicability (only on networks with a combination of dense and conv layers).
> > As mentioned by one of the other reviewers, the insight into why this approach would work better is not provided.
> > For this reason, I do not think the current manuscript can be accepted.

---

### Official Review · AnonReviewer1 · 2018-11-04
**The contribution is limited**

**Rating:** 4
**Confidence:** 4

**Review:**

This paper targets at developing new DFA method to replace BP for neural network model optimization, in order to speed up the training process. The paper is generally written clearly and relatively easy to follow.

My main concern is about significance of the contribution of this paper.

1. the novelty is limited. This paper only simply combines two well-known approach BP and DFA together.

2. performance contribution seems not significant from the proposed approach. In the implementation, the authors only apply their approach to optimize a few top layers. A majority of the layers in the NN model are still optimized via BP.

3. the authors should provide more evaluations on different NN backbones and datasets, to make the experiments stronger and more convincing.

---

> ### Author Response · Authors · 2018-11-13
> **Thank you for valuable comments**
>
> Thanks to your valuable comments, we can improve the quality of the manuscript more than before.
>
> Before answering your main concerns, the manuscript was revised as follow.
>
> 1) The contents of the 2nd paragraph was gradient vanishing problem but it is replaced with the overfitting problem to emphasize the effect of the CDFA.
>
> 2) We made the figures smaller to secure more manuscript spaces.
>
> 3) Minor typos in the manuscripts were revised.
>
> Followings are our response to your comments.
>
> 1) Limited Novelty
> As you pointed out, we suggest the combination of two well-known training methods, BP and binarized DFA. Although the proposed solution is simple, the effect of the training is powerful especially when the training dataset is small. To emphasize the effect of the algorithm, we added the simulation of data augmentation (table 2).
>
> 2) Performance contribution
> Reduced amount of computation by BDFA can be much smaller when we include the computations due to the covolutional layers. Therefore, computational efficiency caused by BDFA can be larger only for the specific NN models. For example, MDNet (referred in the paper) uses only FC fine-tuning, and BDFA can improve the computational efficiency significantly.
> However, when the BDFA is applied instead of BP, we can reduce the required size of the dataset to achieve same test accuracy. Finally, training with the small dataset can additionally improve the effectiveness of the BDFA based training.
>
> 3) Limited evaluations
> We are really sorry that we could not add the evaluations with the different NN models and datasets. Our computation resources are limited so, it is hard to get the results before the deadline. Instead, we added the simulation results with data augmentation, and convolution layer only training. We are still evaluating the more various network configurations and datasets.

---

### Meta-Review · Area_Chair1 · 2018-12-13
**Great explanation of prior work, but limited applicability and no insight or analysis**

**Confidence:** 5
**Recommendation:** Reject

**Metareview:**

This paper proposes a training algorithm for ConvNet architectures in which the final few layers are fully connected.  The main idea is to use direct feedback alignment with carefully chosen binarized (±1) weights to train the fully connected layers and backpropagation to train the convolutional layers. The binarization reduces the memory footprint and computational cost of direct feedback alignment, while the careful selection of feedback weights improves convergence. Experiments on CIFAR-10, CIFAR-100, and an object tracking task are provided to show that the proposed algorithm outperforms backpropagation, especially when the amount of training data is small. The reviewers felt that the paper does a terrific job of introducing the various training algorithms --- backpropagation, feedback alignment, and direct feedback alignment --- and that the paper clearly explained what the novel contributions were. However, the reviewers felt the paper had limited novelty because it combines ideas that were already known, that it has limited applicability because it will not work with fully convolutional architectures, that the baselines in the experiments were somewhat weak, and that the paper provided no insights on why the proposed algorithm might be better than backpropagation in some cases. Regrettably, only one reviewer (R2) participated in the discussion, though this was the reviewer who provided the most constructive review. The AC read the revised paper, and agrees with R2's concerns about the limited applicability of the proposed algorithm and lack of insight or analysis explaining why the proposed training algorithm would improve over backpropagation.